Diets with and without edible cricket support a similar level of diversity in the gut microbiome of dogs

http://orcid.org/0000-0002-8602-7348 Jarett Jessica K. 1
Carlson Anne 2
Rossoni Serao Mariana 3
Strickland Jessica 4
Serfilippi Laurie 4
Ganz Holly H. 1 holly@animalbiome.com
1 AnimalBiome , Oakland, CA , USA
2 Jiminy’s , Berkeley, CA , USA
3 Department of Animal Sciences, Iowa State University , Ames, IA , USA
4 Summit Ridge Farms , Susquehanna, PA , USA
Harrison Xavier
Electronic publication date: 2019 Sep 10
Publication date: 2019
Volume: 7
Electronic Location ID: e7661
Received 2019 Apr 18; Accepted 2019 Aug 12
Copyright: © 2019 Jarett et al.
Copyright year: 2019
Copyright holder: Jarett et al.
License: This is an open access article distributed under the terms of the Creative Commons Attribution License, which permits unrestricted use, distribution, reproduction and adaptation in any medium and for any purpose provided that it is properly attributed. For attribution, the original author(s), title, publication source (PeerJ) and either DOI or URL of the article must be cited.
License URL: https://creativecommons.org/licenses/by/4.0/

Keywords: Gut microbiome, Dog, Edible cricket, Prebiotic, Diet

Funding: Jiminy’s provided financial support for costs related to the diet trial AnimalBiome provided financial support for sequence data analysis Jiminy’s and AnimalBiome provided financial support for sequencing Anne Carlson of Jiminy’s and Jessica Strickland designed the study protocol Jiminy’s and AnimalBiome decided to prepare and publish a manuscript AnimalBiome employees analyzed the data and prepared the manuscript Jiminy’s provided financial support for costs related to the diet trial, AnimalBiome provided financial support for sequence data analysis, and both companies provided financial support for sequencing. Anne Carlson of Jiminy’s and Jessica Strickland designed the study protocol. Jiminy’s and AnimalBiome decided to prepare and publish a manuscript, and AnimalBiome employees analyzed the data and prepared the manuscript. The funders had no role in study design, data collection and analysis, decision to publish, or preparation of the manuscript.

==============================
The gut microbiome plays an important role in the health of dogs. Both beneficial microbes and overall diversity can be modulated by diet. Fermentable sources of fiber in particular often increase the abundance of beneficial microbes. Banded crickets (Gryllodes sigillatus) contain the fermentable polysaccharides chitin and chitosan. In addition, crickets are an environmentally sustainable protein source. Considering crickets as a potential source of both novel protein and novel fiber for dogs, four diets ranging from 0% to 24% cricket content were fed to determine their effects on healthy dogs’ (n = 32) gut microbiomes. Fecal samples were collected serially at 0, 14, and 29 days, and processed using high-throughput sequencing of 16S rRNA gene PCR amplicons. Microbiomes were generally very similar across all diets at both the phylum and genus level, and alpha and beta diversities did not differ between the various diets at 29 days. A total of 12 ASVs (amplicon sequence variants) from nine genera significantly changed in abundance following the addition of cricket, often in a dose-response fashion with increasing amounts of cricket. A net increase was observed in Catenibacterium, Lachnospiraceae [Ruminococcus], and Faecalitalea, whereas Bacteroides, Faecalibacterium, Lachnospiracaeae NK4A136 group and others decreased in abundance. Similar changes in Catenibacterium and Bacteroides have been associated with gut health benefits in other studies. However, the total magnitude of all changes was small and only a few specific taxa changed in abundance. Overall, we found that diets containing cricket supported the same level of gut microbiome diversity as a standard healthy balanced diet. These results support crickets as a potential healthy, novel food ingredient for dogs.

Introduction

The dense, complex community of microbes occupying the gut has been linked to many aspects of health and disease in animals including nutrient absorption (Smith et al., 2013), immune system modulation (Xu et al., 2015), obesity (Rosenbaum, Knight & Leibel, 2015; Marotz & Zarrinpar, 2016), and inflammatory bowel diseases (Willing et al., 2010; Clemente et al., 2012). Alterations in the gut microbiome can occur due to dietary interventions (Flint et al., 2015; Campbell et al., 2016), antibiotic therapy, or fecal microbiome transplantation (Gough, Shaikh & Manges, 2011). Targeting the gut microbiome has become a promising approach not only for treatment of some conditions but also for prevention of disease and improving overall health. Much of this research has focused on humans and on mice as model systems, but dogs (Canis lupus familiaris) also suffer from many of the same health issues, particularly obesity (German et al., 2018), atopic dermatitis, and inflammatory bowel diseases. Dog microbiomes are actually more similar to human microbiomes than mouse microbiomes are in terms of both taxonomic composition and functional gene content (Coelho et al., 2018), and dogs live in the same environment alongside humans, so research on their microbiomes not only contributes to good pet stewardship but also provides a valuable comparative model bridging mice and humans.

The gut microbiome of healthy dogs consists primarily of the phyla Bacteroidetes, Fusobacteria, Firmicutes, and Proteobacteria, and at the genus level is often dominated by Clostridium, Ruminococcus, Dorea, Roseburia, Fusobacterium, Lactobacillus, and others (Suchodolski, Camacho & Steiner, 2008; Middelbos et al., 2010; Handl et al., 2011; Swanson et al., 2011; Li et al., 2017). The abundance of specific microbes from the gut has been linked to health problems in dogs, including obesity (Handl et al., 2013), inflammatory bowel disease (Suchodolski et al., 2012; Honneffer, Minamoto & Suchodolski, 2014), and diarrhea (Bell et al., 2008; Jia et al., 2010). Connections are also beginning to be drawn for atopic dermatitis (Craig, 2016). Given these links and similar associations in humans and mice, much research has focused on promoting or restoring a healthy microbiome in order to improve health. Fecal microbiome transplantation frequently leads to dramatic improvements in health (Gough, Shaikh & Manges, 2011; Xu et al., 2015), but achieving similar results with less invasive approaches is an active area of research.

Dietary modifications are often used to modulate the gut microbiome. Typical approaches include changing ratios of macronutrients (protein, fat, carbohydrate) (Flint et al., 2015; Singh et al., 2017); probiotics, living microorganisms which confer health benefits (Grześkowiak et al., 2015); and prebiotics, fibers or fiber-rich ingredients intended to promote the growth of beneficial bacteria already existing in the gut (Scott et al., 2015). In dogs, the microbiome changes in response to dietary macronutrient ratios (Li et al., 2017; Herstad et al., 2017; Kim et al., 2017). Beet pulp and potato fiber are two prebiotic foods that have been tested in dogs and found to induce changes in microbiome composition at the phylum level, increasing the abundance of Firmicutes and decreasing that of Fusobacteria (Middelbos et al., 2010; Panasevich et al., 2015). The addition of other prebiotics including inulin, fructo-oligosaccharides, and yeast cell wall extract resulted in more minor changes to a few families and genera, but with little overlap in the identity of these taxa between prebiotic sources (Beloshapka et al., 2013; Garcia-Mazcorro et al., 2017). Beans, which combine a possible prebiotic and novel protein source for dogs, affected the abundance of only a few genera (Kerr et al., 2013; Beloshapka & Forster, 2016).

Most prebiotics originate from plant fibers, but the chitinous exoskeletons of arthropods also contain fiber, and many crustaceans and insects are commonly consumed as food items (Rumpold & Schlüter, 2013; Ibitoye et al., 2018). In this study, we focused on banded crickets (Gryllodes sigillatus) and how their use as a protein source in food affected the composition of the microbiome of healthy dogs. There is interest in crickets as a more sustainable protein source for commercial pet food. Cricket production uses substantially less water, land, and feed (De Prins, 2014) and emits far less greenhouse gas (Oonincx et al., 2010) per kilogram than production of other animals such as cows, pigs, and chickens. As a food, crickets contain all essential amino acids (Belluco et al., 2013), as well as minerals, vitamins, fatty acids (Rumpold & Schlüter, 2013), and fiber in the forms of chitin (4.3–7.1% of dry weight) and chitosan (2.4–5.8% of dry weight) (Ibitoye et al., 2018). Chitin is not digestible by dogs, but chitosan is (Okamoto et al., 2001), and to our knowledge its effects on the canine microbiome have not been studied.

Chitosan appears to have a therapeutic effect on diabetes in rats and obesity in mice (Prajapati et al., 2015; Xiao et al., 2016). These health effects were partially modulated via changes in the gut microbiome, although the composition of the community did not always change in the same way (Mrázek et al., 2010; Koppová, Bureš & Simůnek, 2012; Stull et al., 2018). Chitosan and chitin also have potent direct effects on the immune system (Prajapati et al., 2015; Xiao et al., 2016). Chitosan and chito-oligosaccharides increased the concentration of short-chain fatty acids in the gut, which have numerous gut-related and systemic health benefits (Yao & Chiang, 2006; Kong et al., 2014; Koh et al., 2016). In humans, consumption of chitosan did not affect Bifidobacterium abundance (Mrázek et al., 2010), but whole cricket meal (from house crickets, Acheta domesticus) acted as a prebiotic, inducing a 5-fold increase in Bifidobacterium animalis along with a reduction in markers of systemic inflammation (Stull et al., 2018). Together, all of these results suggest that crickets and the fermentable fibers they contain might be beneficial for the gut microbiome.

Whole crickets are a nutritious food that represents both a novel protein and a novel fiber source for dogs. We sought to determine if they increased the abundance of potentially beneficial bacteria, as well as their overall effect on the canine gut microbiome. We used a longitudinal dose-response design with four different diets and 32 animals over a period of 30 days, assessing the microbiome with high-throughput sequencing of the 16S rRNA gene from fecal samples.

Materials and Methods

Animals, diet, and experimental design

In total, 32 male and female beagles from 1 to 8 years of age, median initial weight of 9.68 kg (±1.90 kg) (males 10.97 ± 1.35 kg, females 8.04 ± 1.48 kg), all in apparent good health, non-lactating, and non-pregnant, were used in this study. Dogs were housed in individual pens at the same facility for the duration of the study. The study protocol was reviewed and approved prior to implementation by the Summit Ridge Farms’ Institutional Animal Care and Use Committee and was in compliance with the Animal Welfare Act (approval reference JMYDIGC00118).

A longitudinal dose-response study design was used, with four groups consisting of n = 8 dogs each. Dogs were equally distributed into groups by age, sex, and weight. Dogs were weighed weekly and feeding amounts were adjusted in order to maintain body weight (Table S1). At the initiation of the study, all dogs were switched from their previous diet to a simple, nutritionally complete and balanced base diet with either 0%, 8%, 16%, or 24% of the protein content replaced with whole cricket meal, which they received once a day for the remainder of the study. The remainder of the protein content in the diets was provided by chicken. Cricket meal was produced by Entomo Farms in accordance with current good manufacturing practice and the Food Safety Manufacturing Act, and extruded kibble was custom produced for this study by Wenger. Nutritional analysis of each diet is shown in Table S2. The 0% cricket meal diet was considered the control diet. Fecal samples were collected from each dog one day before diet was initially switched and again at 14 and 29 days thereafter. There were no gastrointestinal issues (diarrhea, gas, vomiting, etc.) reported by the study facility during this study, no significant percent changes in weights of animals from day 0 to day 29 in any diet group (one sample t-tests, P > 0.10), and no significant differences in percent changes in weights of animals between diets (one-way ANOVA, F(3, 28) = 0.291, P = 0.832).

Sample collection, processing, and sequencing

Samples were collected and stored at 4–8 °C in two ml screw cap tubes containing 70% ethanol and silica beads. Fecal material was isolated from preservation buffer by pelleting (centrifugation at 10,000×g for 5 min, pouring off supernatant), then genomic DNA was extracted using the 100-prep DNeasy PowerSoil DNA Isolation Kit (Qiagen). Samples were placed in bead tubes containing C1 solution and incubated at 65 °C for 10 min, followed by 2 min of bead beating, then following the manufacturer’s protocol. Amplicon libraries of the V4 region of the 16S rRNA gene (505F/816R) were generated using a dual-indexing one-step PCR with complete fusion primers (Ultramers, Integrated DNA Technologies, Coralville, IA, USA) with multiple barcodes (indices), adapted for the MiniSeq platform (Illumina, San Diego, CA, USA) from Pichler et al. (2018). PCR reactions containing 0.3-30 ng template DNA, 0.1 μl Phusion High-Fidelity DNA Polymerase (Thermo Fisher, Waltham, MA, USA), 1X HF PCR Buffer, 0.2 mM each dNTP, and 10 μM of the forward and reverse fusion primers were denatured at 98 °C for 30 s, cycled 30 times at 98 °C, 10 s; 55 °C, 30 s; 72 °C, 30 s; incubated at 72 °C for 4 min 30 s for a final extension, then held at 6 °C. PCR products were assessed by running on 2% E-Gels with SYBR Safe (Thermo Fisher, Waltham, MA, USA) with the E-Gel Low Range Ladder (Thermo Fisher, Waltham, MA, USA), then purified and normalized using the SequalPrep Normalization Kit (Thermo Fisher, Waltham, MA, USA) and pooled into the final libraries, each containing 95 samples and one negative control. The final libraries were quantified with QUBIT dsDNA HS assay (Thermo Fisher, Waltham, MA, USA), diluted to 1.5 pM and denatured according to Illumina’s specifications for the MiniSeq. Identically treated phiX was included in the sequencing reaction at 25%. Paired-end sequencing (150 bp) was performed on one mid-output MiniSeq flow cell (Illumina, San Diego, CA, USA) per final library. All sequences and corresponding metadata are freely available in the NCBI Sequence Read Archive (PRJNA525542).

Alpha and beta diversity analysis

Sequence data were analyzed with QIIME2, a plugin-based microbiome analysis platform (version 2018.4.0). After demultiplexing, the q2-dada2 plugin (Callahan et al., 2016) was used to perform quality filtering and removal of phiX, chimeric, and erroneous sequences, and identify amplicon sequence variants (ASVs) (i.e., 100% nucleotide identity). ASVs were classified with classify-sklearn in the q2-feature-classifier plugin (Garreta & Moncecchi, 2013; Bokulich et al., 2018b), using a classifier trained on the 515-806 region of the SILVA reference database (version 132) (Quast et al., 2013) (Data S1). A two-way ANOVA of sampling depth with cricket protein and sampling day as factors was performed in R, followed by pairwise Welch’s t-tests with Bonferroni correction of P-values to determine if sampling depths were significantly different between treatment groups or sampling days. One sample with a highly divergent community (top 25 ASVs comprising less than 50% of reads) was removed prior to all downstream analyses, and remaining samples were subsampled to a depth of 26,679 reads for alpha and beta diversity analyses (Data S1).

Several metrics of alpha diversity (Shannon diversity, Pielou’s evenness, observed ASVs) were calculated in QIIME2 (hereafter, version 2018.11.0) with the q2-diversity plugin, and the ratio of Firmicutes to Bacteroidetes (F:B ratio) was calculated in R. Differences between day 0 and day 29 were assessed for each response variable using pairwise-differences from the q2-longitudinal plugin (Bokulich et al., 2018a) with cricket protein as a fixed effect. A linear mixed-effects model was also fitted to each variable with linear mixed-effects (q2-longitudinal) with cricket protein as a fixed effect and random intercepts for individual animals. Alpha diversity values and F:B ratio were exported and plotted in R with ggplot2 (Gómez-Rubio, 2017).

Changes in beta diversity between day 0 and day 29, as measured by Bray–Curtis dissimilarity, were tested with pairwise distances (q2-longitudinal) with cricket protein as a fixed effect. The pairwise distances between day 0 and day 14, and between day 0 and day 29, were used as input to linear mixed-effects with cricket protein as a fixed effect and random intercepts for individual animals. Principal coordinates analysis (PCoA) was performed and visualized with phyloseq in R (McMurdie & Holmes, 2013).

Differential abundance analysis

Three approaches were used to detect ASVs that differed between diets. First, for longitudinal analysis of composition of microbiomes (ANCOM) (Mandal et al., 2015; Mandal, 2018), ASVs with fewer than 500 total reads or found in less than 10% of samples were removed from the unrarefied data. The use of log ratios in ANCOM essentially normalizes for sequencing depth (Mandal et al., 2015), and this approach has been demonstrated to be robust to differences in library sizes (Weiss et al., 2017). Cricket protein was modeled as a main variable, individual animals were modeled as random effects, and a w0 cutoff of 0.7 (where larger values are more conservative) and P-value of 0.05 were applied. The second approach used feature-volatility from q2-longitudinal. This machine learning pipeline determines which features (here, ASVs) were most predictive of the time point that a sample was collected, i.e., which ASVs were changing the most over the time course of the experiment. Unrarefied data was used, with ASVs occurring in fewer than four samples or with fewer than 200 total reads removed. Like ANCOM, the random forest algorithm implemented in q2-longitudinal is robust to differences in library size, so we chose to use unrarefied data. Feature-volatility was run with state (day) and individual animal identifiers, 5-fold cross-validation, the random forest regressor for sample prediction, and 1,000 trees for estimation. This step was run 10 times, and the top 10 most important features (ASVs) from each run were retained. All ASVs which appeared in the results at least twice (n = 15) were tested individually by building a model with linear mixed-effects, with parameters as previously described. For ASVs with significant (P ≤ 0.05) interactions of diet and time, the relative abundances in different diets at day 29 were compared with pairwise Wilcoxon rank-sum tests in the coin package (Hothorn et al., 2008). False discovery rate was controlled at 10% by the q-value package (Storey et al., 2017), and comparisons with P ≤ 0.05 and q ≤ 0.1 were considered significant. Lastly, we calculated the difference in relative occurrence metric (DIROM) (Ganz et al., 2017) for each ASV on day 29. The rarefied ASV table was transformed to a presence/absence matrix, and dogs were divided into two groups, those eating the control diet (n = 8) and those eating any percentage of cricket (n = 24). The DIROM of each ASV in the control and cricket groups was calculated, and only ASVs with a DIROM of at least |0.3| (n = 17) were retained. The abundance of each ASV at day 29 in control and all cricket diets combined was compared with Wilcoxon rank-sum pairwise tests and q-values as described above.

Results

To determine the effects of cricket consumption on the gut microbiome of dogs, we sequenced a total of 8,788,186 reads from 96 samples, resulting in 26,679 to 139,682 non-chimeric reads per sample (Table S3) and a total of 536 ASVs. One sample (Dog ID 13603, day 14) was dropped from downstream analyses due to a highly divergent community. In this sample, the 25 ASVs with the highest overall abundance across the dataset comprised less than 50% of reads. A 2-way ANOVA on sampling depth and pairwise Welch’s t-tests revealed significantly fewer reads in samples from day 0 than day 29 (t(62) = −3.2552, P = 0.002) (Table S4; Fig. S1). Because of this, we rarefied the data for most analyses to avoid the possibility of confounding the effects of time and sampling depth.

Microbial communities in all dogs across the course of the study were broadly similar at the phylum level (Fig. S2) and across the top 15 most abundant genera in the dataset (Fig. 1; Fig. S3). Neither Shannon diversity, Pielou’s evenness, nor richness (observed ASVs) changed significantly between day 0 and day 29 in any diet, except for the 8% cricket diet where Shannon diversity increased significantly over time (Table 1; Fig. 2B; Figs. S4 and S5). The magnitude of change in alpha diversity from day 0 to day 29 was not different between diets, for any alpha diversity metric (Kruskal–Wallis tests, Shannon diversity H(3) = 4.946023, P = 0.175; Pielou’s evenness H(3) = 2.036932, P = 0.564; richness H(3) = 3.58158, P = 0.310). Initial alpha diversity values and the rates of change did not differ between diets in linear mixed-effects models (Table S5). Likewise, the ratio of Firmicutes to Bacteroidetes did not change significantly over time (Table 1), and neither the initial F:B ratios, rates of change, or magnitude of change differed between diets (Kruskal–Wallis test, H(3) = 1.707386, P = 0.635; Fig. S6; Table S5). Samples from all time points and diets were intermingled in principal coordinates plots using Bray–Curtis dissimilarity (Fig. 3). Similar to results for alpha diversity, the distances between samples from day 0 and day 29 in Bray–Curtis dissimilarity were not significantly different between diets (Kruskal–Wallis test, H(3) = 1.119318, P = 0.772413). When comparing the distances between day 0 and day 14, or between day 0 and day 29, for each dog using linear mixed-effects models, there were no significant effects of time or diet (Table S5).

Figure 1 Bacterial community composition at the genus level in dogs eating control diets and diets containing cricket is similar.

Genus-level composition of gut microbiomes in dogs consuming diets containing different amounts of cricket meal, averaged across eight dogs per diet and sampled longitudinally at (A) day 0, (B) day 14, and (C) day 29. Only the 15 most abundant genera are shown.

Figure 2 Alpha diversity is similar in dogs eating control diets and diets containing cricket.

Shannon diversity of gut microbiomes of dogs consuming diets containing (A) 0% cricket meal, (B) 8% cricket meal, (C) 16% cricket meal, or (D) 24% cricket meal, averaged across eight dogs per diet and sampled longitudinally over the course of 29 days.

Figure 3 Beta diversity of bacterial communities in dogs eating control diets and diets containing cricket does not differ.

Principal coordinates analysis of Bray–Curtis dissimilarity of gut microbiomes of dogs consuming diets containing different amounts of cricket meal, sampled longitudinally over the course of 29 days.

Table 1 Wilcoxon signed-rank tests of changes in alpha diversity in paired samples from day 0 and day 29, and mean values at day 0 and day 29.

	Treatment group	W (Wilcoxon signed-rank test)	P-value	FDR P-value	Mean, day 0	Mean, day 29	
Shannon diversity	0% Cricket	9	0.208	0.415	4.7824	4.9220	
8% Cricket	0	0.012	0.047	4.5962	5.0157	
16% Cricket	14	0.575	0.575	4.7651	4.8284	
24% Cricket	14	0.575	0.575	4.7098	4.8459	
Pielou’s evenness	0% Cricket	12	0.401	0.534	0.7305	0.7454	
8% Cricket	6	0.093	0.372	0.7094	0.7453	
16% Cricket	12	0.401	0.534	0.7344	0.7453	
24% Cricket	17	0.889	0.889	0.7371	0.7487	
Richness (observed ASVs)	0% Cricket	11	0.327	0.654	94.3750	97.8750	
8% Cricket	3	0.063	0.252	91.0000	107.2500	
16% Cricket	13.5	0.932	0.932	90.1250	89.6250	
24% Cricket	15	0.673	0.897	85.8750	91.2500	
Firmicutes:Bacteroidetes ratio	0% Cricket	7	0.123	0.165	0.7372	1.0414	
8% Cricket	11	0.327	0.327	1.0080	1.3063	
16% Cricket	5	0.069	0.137	0.7816	1.6184	
24% Cricket	2	0.025	0.100	0.8154	1.7072	
Note:

False discovery rate (FDR) corrected P-values in bold are considered statistically significant.

Three methods were used to assess differences in abundance of specific ASVs between diets, and yielded different results. Longitudinal ANCOM did not find any ASVs that significantly differed in abundance between diets (Table S6). With the machine learning-based feature-volatility approach, we identified 15 ASVs that occurred at least twice in the 10 iterations. Four of these ASVs had an interaction effect of time and diet that was significant when tested with a linear mixed-effects model (Table S7). Among these, three differed significantly in abundance between diets in Wilcoxon rank-sum tests. Two Catenibacterium sp. ASVs differed between diets, but had opposite patterns. Catenibacterium sp. 2 (numbers were arbitrarily assigned to distinguish ASVs) was significantly less abundant in the 24% cricket diet than in control or 8% cricket (Table 2; Fig. 4A), and Catenibacterium sp. 1 was significantly more abundant in the 24% cricket diet relative to control and 8% cricket (Table 2; Fig. 4B). Lachnospiraceae [Ruminococcus] torques sp. was significantly increased in the 24% cricket diet relative to control (Fig. 4C; Table 2).

Figure 4 Three ASVs differ in abundance between dogs eating control diets and diets containing cricket.

Relative abundance at day 29 of ASVs identified by feature-volatility from q2-longitudinal: (A) Catenibacterium sp. 2, (B) Catenibacterium sp. 1, and (C) [Ruminococcus] torques sp. Significant differences between diets are indicated with an asterisk (Wilcoxon rank-sum tests, P ≤ 0.05, q ≤ 0.1). Numbers denote different ASVs from the same genus and are arbitrary.

Table 2 Wilcoxon rank-sum tests of the abundance in different diets of ASVs detected by longitudinal feature-volatility.

Genus	Comparison	U	P-value	FDR q-value	Feature ID	
Catenibacterium sp. 2	0% vs. 24%	−2.9428	0.001	0.015	8e83238a1a628f1db6f17d9e5524714f	
Catenibacterium sp. 1	0% vs. 24%	−2.8356	0.003	0.015	1541faf3a457cc8cc05b01ce30983449	
Catenibacterium sp. 1	8% vs. 24%	−2.7305	0.006	0.021	1541faf3a457cc8cc05b01ce30983449	
Catenibacterium sp. 2	8% vs. 24%	−2.6255	0.007	0.021	8e83238a1a628f1db6f17d9e5524714f	
[Ruminococcus] torques group sp.	0% vs. 24%	2.1020	0.041	0.093	12615dfed222d35c1582cbd6cef48013	
Catenibacterium sp. 1	16% vs. 24%	−1.7854	0.081	0.119	1541faf3a457cc8cc05b01ce30983449	
Catenibacterium sp. 2	0% vs. 16%	−1.7867	0.083	0.119	8e83238a1a628f1db6f17d9e5524714f	
Catenibacterium sp. 2	16% vs. 24%	−1.7854	0.083	0.119	8e83238a1a628f1db6f17d9e5524714f	
[Ruminococcus] torques group sp.	8% vs. 24%	1.6816	0.102	0.122	12615dfed222d35c1582cbd6cef48013	
[Ruminococcus] torques group sp.	0% vs. 16%	1.6803	0.108	0.122	12615dfed222d35c1582cbd6cef48013	
Blautia sp.	0% vs. 24%	1.5753	0.125	0.122	2a7169b7465789a82b4f47c3d934d259	
Catenibacterium sp. 1	0% vs. 16%	−1.5753	0.128	0.122	1541faf3a457cc8cc05b01ce30983449	
[Ruminococcus] torques group sp.	16% vs. 24%	1.3663	0.178	0.157	12615dfed222d35c1582cbd6cef48013	
Blautia sp.	0% vs. 16%	1.2603	0.235	0.193	2a7169b7465789a82b4f47c3d934d259	
Catenibacterium sp. 2	8% vs. 16%	−1.1552	0.275	0.211	8e83238a1a628f1db6f17d9e5524714f	
Blautia sp.	0% vs. 8%	1.0502	0.329	0.236	2a7169b7465789a82b4f47c3d934d259	
[Ruminococcus] torques group sp.	8% vs. 16%	0.9452	0.379	0.250	12615dfed222d35c1582cbd6cef48013	
Catenibacterium sp. 2	0% vs. 8%	−0.9480	0.392	0.250	8e83238a1a628f1db6f17d9e5524714f	
Catenibacterium sp. 1	0% vs. 8%	−0.7882	0.469	0.283	1541faf3a457cc8cc05b01ce30983449	
Catenibacterium sp. 1	8% vs. 16%	−0.6301	0.575	0.330	1541faf3a457cc8cc05b01ce30983449	
Blautia sp.	16% vs. 24%	0.5251	0.650	0.356	2a7169b7465789a82b4f47c3d934d259	
[Ruminococcus] torques group sp.	0% vs. 8%	0.4201	0.723	0.377	12615dfed222d35c1582cbd6cef48013	
Blautia sp.	8% vs. 16%	−0.3151	0.790	0.394	2a7169b7465789a82b4f47c3d934d259	
Blautia sp.	8% vs. 24%	−0.1050	0.957	0.458	2a7169b7465789a82b4f47c3d934d259	
Note:

False discovery rate (FDR) q-values indicate the estimated false discovery rate if a given test is considered significant. P-values in bold are considered significant.

Lastly, there were 17 ASVs with a DIROM of at least 30% between control diets and all cricket diets combined, none of which were detected with the other methods (Table 3). When comparing the abundance of these ASVs in control diets and all cricket diets, six were significantly more abundant in controls, two were more abundant in cricket diets, and nine were not significantly different between diets (Fig. 5; Table 3). For Catenibacterium sp. 3, Collinsella sp., Faecalitalea sp., and Lachnospiraceae NK4A136 group, an approximate dose-response relationship was observed between ASV abundance and the amount of cricket in the diet (Figs. 5D–5H).

Figure 5 Nine ASVs differ in both relative occurrence and abundance between dogs eating control diets and diets containing cricket.

Relative abundance at day 29 of ASVs with at least 30% DIROM and significant differences between control diet and all cricket diets combined (Wilcoxon rank-sum tests, P ≤ 0.05, q ≤0.1). ASVs shown are (A) Bacteroides sp. 1, (B) Bacteroides sp. 2, (C) Candidatus Arthromitus sp., (D) Catenibacterium sp. 3, (E) Collinsella sp., (F) Faecalibacterium sp., (G) Faecalitalea sp., (H) Lachnospiraceae NKA136 group sp., (I) Megamonas sp. Numbers denote different ASVs from the same genus and are arbitrary.

Table 3 Wilcoxon rank-sum tests of the abundance in control and cricket-containing diets of ASVs detected by DIROM (>0.3).

Genus	DIROM	U	P-value	FDR q-value	Feature ID	
Candidatus Arthromitus sp.	0.3333	2.5368	0.011	0.046	91ed4b4dcf2572f842aaabd01b8fbaaa	
Bacteroides sp. 2	0.3333	2.4188	0.013	0.046	3d6657c33fee3a6c6ea2b90982a59c1a	
Faecalitalea sp.	0.4583	−2.4068	0.014	0.046	f7870a4e6cccb8a029cf6f0091c106f5	
Catenibacterium sp. 3	0.4167	−2.3637	0.015	0.046	7a21bb7da3ce9ff23c72fc82d270bd56	
Faecalibacterium sp.	0.4167	2.3565	0.016	0.046	a383bbf0dc0a9f5c17cf3303b55af028	
Lachnospiraceae NK4A136 group sp.	0.3333	2.4561	0.016	0.046	b05080b88bcf581a4e0ad0be14acecdb	
Megamonas sp.	0.3333	2.3854	0.033	0.074	f21ae878ede3c6f26bd559bb098195f4	
Collinsella sp.	0.4167	2.1137	0.035	0.074	9a67e1965fe9ce8fed8d8fbe40c3ec36	
Bacteroides sp. 1	0.3333	1.9536	0.049	0.092	f243876948ab4496310b934dcf6b17d3	
Blautia sp.	0.3333	−1.8311	0.119	0.148	02b4317d07dfa2d4bcf0b885431ba6ce	
Ruminococcaceae uncultured bacterium	0.3333	−1.8311	0.122	0.148	f2047101a7ec62adb3418e40ecef8eb2	
Prevotella 9 sp.	0.3333	−1.8311	0.128	0.148	da6a7cba87e0895b9cbe6037b9bd8b3b	
Faecalibacterium sp.	0.3333	−1.8311	0.129	0.148	78d4f442773fc72402c01693d81a45e6	
Clostridium sensu stricto 1 sp.	0.3333	−1.8311	0.130	0.148	1d49df56b83a383d8b998af9d148f52c	
Catenibacterium sp. 4	0.3333	−1.5350	0.130	0.148	e52cbb022d6df967d43c6019a71eaa2b	
Lachnospiraceae uncultured bacterium	0.3333	1.4877	0.154	0.164	d7354463af117a55b599229db7109b1d	
Anaeroplasma sp.	0.3333	0.6140	0.544	0.544	ca35d5f5f1a0f225e7f1b39a00dff592	
Note:

False discovery rate (FDR) q-values indicate the estimated false discovery rate if a given test is considered significant. P-values in bold are considered significant.

Discussion

To date, cricket is not widely available in dog food, so it could be a novel protein and fiber source for most dogs. Here, we assessed its effect on the diversity and composition of the gut microbiome. Based on a recent study of cricket consumption in humans (Stull et al., 2018), we predicted that the diversity and overall composition of the community would not change in dogs eating cricket, and that only a few taxa would differ significantly in relative abundance between different diets.

At a community level, diversity metrics showed no significant differences between diets containing different amounts of cricket. Alpha diversity metrics did not change over time, except Shannon diversity in 8% cricket increased between day 0 and day 29 (Table 1), however this treatment group also began the study with the lowest Shannon diversity (Fig. 2). The ratio of Firmicutes to Bacteroidetes also did not differ between diets or change over time (Fig. S6; Table S5). Increases in this metric have been linked to obesity in humans and mice (Rosenbaum, Knight & Leibel, 2015), but more recent meta-analyses have called this association into question (Sze & Schloss, 2016). Finally, beta diversity showed no differences or clustering due to diet (Fig. 3; Table S5), meaning that the community composition was not shifted in any consistent manner by cricket diets. Overall, the level of diversity supported by cricket diets is the same as that of a healthy balanced diet without cricket. These results are similar to those of Stull et al. (2018) where alpha and beta diversity in human gut microbiomes were unaltered by cricket consumption.

In agreement with our predictions, only a few ASVs within the gut microbiome changed significantly in abundance due to cricket diets. None of these changes were of sufficient magnitude to be statistically significant in a standard differential abundance analysis (ANCOM, Table S6), so we pursued two alternative approaches. The first of these (q2-longitudinal analysis) detected ASVs with the greatest change in abundance over the time of the study, and the second (DIROM) focused on those that differed the most in occurrence between diets at the study endpoint. Following initial detection by DIROM, only those ASVs which also differed significantly in abundance between the control diet and all cricket diets combined at the endpoint (day 29) were retained. In combination, these two methods may give a more complete picture of changes occurring in the canine gut microbiome.

In total, 12 ASVs differed in abundance between diets, three of which were detected with longitudinal analysis and nine with DIROM. Four ASVs increased and three decreased in a dose-response fashion with cricket content of the diet, so while overall changes in abundance were small, these trends with increasing amounts of cricket lend credence to the results (Figs. 4 and 5). Five other ASVs displayed a pattern of greater abundance in the control diet, and reduction to a lower, approximately equal level in all cricket diets. These included Bacteroides sp. 1 and 2, Candidatus Arthromitus sp., Faecalibacterium sp., and Megamonas sp. (Figs. 5A–5C, 5F, 5I). Two ASVs that had low overall prevalence at day 29 (Candidatus Arthromitus sp., Megamonas sp.) are not discussed further.

ASVs that increased in abundance included genera with positive and negative connotations for health. A Lachnospiraceae [Ruminococcus] torques group sp. ASV increased significantly between the control diet and 24% cricket (Fig. 4C; Table 2). This group is known to degrade mucin and produce butyrate (Hoskins et al., 1992; Crost et al., 2013; Takahashi et al., 2016), and along with Lachnospiraceae [Ruminococcus] gnavus has recently been re-classified as genus Blautia (Liu et al., 2008; Lawson & Finegold, 2014). Excessive mucin degradation by gut bacteria can lead to a loss of the mucus layer that protects host cells from direct contact with bacteria, leading to destabilization of the intestinal barrier and inflammation (Hoskins et al., 1992; Breban et al., 2017; Van Herreweghen et al., 2018). In humans, it is more abundant in inflammatory bowel disease (Png et al., 2010; Hall et al., 2017) and is enriched by diets low in FODMAPs (fermentable oligo-, di-, and mono-saccharides) (Halmos et al., 2015). However, in dogs a higher relative abundance of Lachnospiraceae [Ruminococcus] was observed when beans were included in the diet (Beloshapka & Forster, 2016). Beans are high in FODMAPs (Fedewa & Rao, 2014), so this suggests that Lachnospiraceae [Ruminococcus] may respond differently to diet in dogs and humans. Notably, this increase in dogs observed by Beloshapka and Forster occurred without any ill effects on health or digestive symptoms (Beloshapka & Forster, 2016).

Three different ASVs of Catenibacterium sp. were affected by cricket diets in different ways, with one decreasing (Catenibacterium sp. 2, Fig. 4A) and the others increasing with higher amounts of cricket (Fig. 4B and 5D; Tables 2 and 3). Both in vitro work and research in cats implicate Catenibacterium in the fermentation of dietary starches (Hooda et al., 2013; Yang et al., 2013), so this increase was consistent with crickets providing increased fiber. Catenibacterium produces short-chain fatty acids including acetate, lactate, and butyrate (Kageyama & Benno, 2000), which have numerous health benefits (Koh et al., 2016), suggesting that the overall increase of Catenibacterium in dogs consuming cricket may be health-promoting. Little functional information is available on Faecalitalea, but they are also capable of producing lactate and butyrate (Kageyama, Benno & Nakase, 1999), so the significantly greater abundance and relative occurrence in dogs eating cricket is a further indication of possible benefit of this diet (Fig. 5G; Table 3).

More ASVs were significantly decreased than increased with cricket diets, and these constituted a larger total decline in relative abundance (Fig. 5). Multiple Bacteroides sp. ASVs showed the same decrease to near-zero abundance in all cricket diets (Figs. 5A and 5B; Table 3). Bacteroides are functionally important members of the gut microbiome, utilizing diverse starches and sugars and modulating the host immune system (Wexler & Goodman, 2017), but they can also cause opportunistic infections and may promote the development of colon cancer (Feng et al., 2015). Generally, they are enriched by high fat, high protein, low-fiber diets (Flint et al., 2015), so decreased abundance and prevalence of Bacteroides sp. ASVs in cricket diets that contain more fiber than the control diet would be expected, and may be beneficial to health. This is particularly important given that many commercially available dog foods are somewhat low in fiber, and may contain less fiber as fed than the guaranteed analysis would suggest (Farcas, Larsen & Fascetti, 2013). A Faecalibacterium sp. ASV displayed a similar trend (Fig. 5F), which may be less beneficial because in dogs low numbers of Faecalibacterium are common in lymphoma (Gavazza et al., 2018). However, decreases in this genus were also observed without negative health effects in dogs eating black beans as a component of the diet (Beloshapka & Forster, 2016), and in dogs eating fresh beef (Herstad et al., 2017). Collinsella is known to increase in humans consuming low-fiber diets (Gomez-Arango et al., 2018), and is frequently increased in inflammatory bowel disease (Swidsinski et al., 2002), so the observed decrease in dogs eating cricket diets with a higher fiber content is both expected and suggestive of a positive effect on health. A Lachnospiraceae NK4A136 group sp. ASV also decreased in cricket diets compared to control (Fig. 5H; Table 3), but the potential impacts of this on dog health are unclear. The Lachnospiraceae family are butyrate producers that are abundant in the gut microbiomes of mammals (Meehan & Beiko, 2014) and generally associated with gut health (Biddle et al., 2013). Reduced abundance of Lachnospiraceae has been found in colorectal cancer, however this family is also very functionally diverse (Seshadri et al., 2018) so it is difficult to draw further conclusions about the health implications of this change in dogs.

The taxa that we observed changing in response to cricket consumption differ from those detected in previous studies in humans, as well as in studies of chitin and chitosan supplementation (Mrázek et al., 2010; Koppová, Bureš & Simůnek, 2012; Zheng et al., 2018; Stull et al., 2018). The composition of the gut microbiome differs in dogs and humans (Swanson et al., 2011), so we expected that the precise taxa that changed would also be different, even though the large-scale patterns in alpha and beta diversity were similar. In humans, cricket increased Bifidobacterium and decreased Lactobacillus and Acidaminococcus, among others (Stull et al., 2018); while dogs in this study did have Lactobacillus in their gut microbiomes (Fig. 1), the abundance did not differ between diets. Bifidobacterium was not abundant in these dogs and most other genera highlighted in Stull et al. were absent. However, both of the main taxa that were enriched by cricket in dogs ([Ruminococcus] torques group, Catenibacterium) are thought to have roles in the fermentation of fiber (Hoskins et al., 1992; Hooda et al., 2013; Yang et al., 2013; Crost et al., 2013). Chitin and chitosan, two types of dietary fiber found in crickets, have also been assessed in isolation for their impact on the gut microbiome but did not have the same effects as whole crickets (Mrázek et al., 2010; Koppová, Bureš & Simůnek, 2012; Zheng et al., 2018). A unifying feature of these studies is that a relatively small number of taxa change in abundance, and the overall composition of the community is minimally affected, which parallels our observations in dogs.

One caveat of the current work is that all dogs had been eating a different diet prior to the initiation of the study, so adaptation to the base diet was occurring during the first weeks of the study. As a result, the abundance of several ASVs changed significantly over time but in the same manner across all diets (data not shown), which may have made genuine differences between diets more difficult to detect. Future studies should include an adaptation period to the control diet prior to the first sampling to minimize the effect of this change and increase the power of longitudinal sampling to detect differences. Assessments of other metrics of health such as immune function, blood glucose, and satiety may reveal further benefits of cricket diets in future studies.

Conclusions

In summary, we tested the effects of diets containing up to 24% cricket on the gut microbiome of domestic dogs. We predicted that changes in the overall composition of the community would be minimal, and that few taxa would change in abundance in response to cricket. We found that cricket diets support the same level of microbial diversity as a standard healthy balanced diet. The alpha and beta diversity of the community did not differ between the control diet and diets containing up to 24% cricket. The addition of cricket resulted in small but statistically significantly changes in the abundance of 12 ASVs from nine genera, but only a few of these ASVs comprised more than 1% of the total community (Figs. 4 and 5), so we hypothesize that the impact of cricket on the functional capacity of the gut microbiome is correspondingly small. However, functional responses involved in health such as short-chain fatty acid concentrations were not measured directly and should be investigated in the future. Cricket diets could also be tested in dogs suffering from inflammatory bowel disease, fiber-responsive diarrhea (Leib, 2000), obesity, and other maladies to assess if this novel fiber and protein source with immune modulating properties (Prajapati et al., 2015; Xiao et al., 2016; Stull et al., 2018) could be beneficial.

Supplemental Information

Supplemental Information 1 Raw and rarefied ASV tables.

The raw ASV table contains one sample which was dropped from all analyses due to a highly divergent community relative to other samples. The rarefied ASV table is subsampled to a depth of 26,679 reads per sample.

Click here for additional data file.

Supplemental Information 2 Sex, age, weight, and weight change for all animals.

Click here for additional data file.

Supplemental Information 3 Nutritional analysis of diets by dry weight.

Click here for additional data file.

Supplemental Information 4 Metadata, read counts, and observed ASVs for all samples.

Click here for additional data file.

Supplemental Information 5 Two-way analysis of variance of read counts.

Click here for additional data file.

Supplemental Information 6 Linear mixed-effects model results for alpha and beta diversity metrics.

Click here for additional data file.

Supplemental Information 7 Analysis of composition of microbiomes at a P-value of 0.05, for ASVs with at least 500 total reads and minimum prevalence of 10% of samples.

Click here for additional data file.

Supplemental Information 8 Linear mixed-effects model results for ASVs with significant interactions between time and diet, identified with feature-volatility approach.

Click here for additional data file.

Supplemental Information 9 Sequencing reads per sample, grouped by date of sampling.

Click here for additional data file.

Supplemental Information 10 Phylum-level composition of gut microbiomes in dogs consuming diets containing different amounts of cricket meal, sampled longitudinally over the course of 29 days.

Low-abundance phyla Euryarchaeota, Synergistetes, Spirochaetes, Deferribacteres are not shown.

Click here for additional data file.

Supplemental Information 11 Genus-level composition of gut microbiomes in dogs consuming diets containing different amounts of cricket meal, sampled longitudinally over the course of 29 days.

Only the 15 most abundant genera are shown.

Click here for additional data file.

Supplemental Information 12 Pielou’s evenness of gut microbiomes of dogs consuming diets containing different amounts of cricket meal, averaged across eight dogs per diet and sampled longitudinally over the course of 29 days.

Click here for additional data file.

Supplemental Information 13 Richness (observed ASVs) of gut microbiomes of dogs consuming diets containing different amounts of cricket meal, averaged across eight dogs per diet and sampled longitudinally over the course of 29 days.

Click here for additional data file.

Supplemental Information 14 Ratio of Firmicutes to Bacteroidetes in gut microbiomes of dogs consuming diets containing different amounts of cricket meal, averaged across eight dogs per diet and sampled longitudinally over 29 days.

Click here for additional data file.

The authors wish to thank K. Copren, S. Redner, and Z. Entrolezo for processing samples, and K. Goodman for initial sequence data analysis. Finally, we acknowledge the important contributions of L. Jarett, T. and D. Carlson, C., D., and Y. Ganzborn, and L. N. Dog.

Additional Information and Declarations

Competing Interests

Author Contributions

Animal Ethics

DNA Deposition

Data Availability

Anne Carlson is the founder and CEO of Jiminy’s, a company that makes dog treats and foods containing edible cricket. Holly H. Ganz and Jessica K. Jarett are employees of AnimalBiome, a company that performs gut microbiome testing for cats and dogs. Jessica Strickland and Laurie Serfilippi are both employed by Summit Ridge Farms.

Jessica K. Jarett analyzed the data, prepared figures and/or tables, authored or reviewed drafts of the paper, approved the final draft.

Anne Carlson conceived and designed the experiments, contributed reagents/materials/analysis tools, approved the final draft.

Mariana Rossoni Serao analyzed the data, approved the final draft.

Jessica Strickland conceived and designed the experiments, performed the experiments, approved the final draft.

Laurie Serfilippi performed the experiments, approved the final draft.

Holly H. Ganz performed the experiments, contributed reagents/materials/analysis tools, approved the final draft.

The following information was supplied relating to ethical approvals (i.e., approving body and any reference numbers):

Summit Ridge Farms’s Institutional Animal Care and Use Committee provided full approval for this research (JMYDIGC00118).

The following information was supplied regarding the deposition of DNA sequences:

The 16S rRNA PCR amplicon sequences are available at the NCBI Sequence Read Archive: PRJNA525542.

The following information was supplied regarding data availability:

The raw ASV table and rarefied ASV table are available in the Supplemental File.

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
