# Peer review of "Diets with and without edible cricket support a similar level of diversity in the gut microbiome of dogs"

_PeerJ, doi:10.7717/peerj.7661_

## Round 0.1 · original submission · Minor Revisions

Your manuscript has now been assessed by two expert reviewers. You will see that both find merit in your paper.

Reviewer 1 in particular has provided a number of helpful comments for your revision. In particular, I agree that the wording of the title should be amended given that you do not assay the function of these microbial communities nor follow up the consequences of dietary manipulation / microbiome shifts for individual health.

Reviewer 1 ·

Basic reporting

1. On line 177 please clarify that ‘100% operational taxonomic unit’ means ‘100% nucleotide identity’ among the region of the 16S rRNA gene you amplified. An ‘operational taxonomic unit’ is another term that needs to be defined so it’s best to just expand here on exactly what you mean and leave out ‘operational taxonomic unit’ considering you never mention this term in the manuscript again.

2. On line 331, please clarify that the lack of ill effects was observed in another study – not this one.

3. Table S6 is referenced on line 263, but a table S6 is not included in the supplementary materials.

4. Some readers may not know that high fat, high protein diets are typically low in fiber. Please expand and include an appropriate reference that this is the case near lines 346-347. This will really emphasize the potential benefits of a diet with more fiber.

5. Please write out inflammatory bowel disorder along with the acronym IBD on line 325 or include the acronym earlier in your manuscript so the reader will know what you are referring too.

6. On line 207, you mention a cutoff of 0.7. Is there a name or variable for what value you are cutting off?

7. On line 323, you describe the Lachnospiraceae [Ruminococcus] torques group is known to degrade mucin. Please expand on why this function connotes either positive or negative health consequences as you state in the intro sentence to this paragraph.

8. Please expand the [Ruminococcus] sp. label in Figure to be [Ruminococcus] torques group sp.. This is how you refer to this group in the text so keeping the labeling consistent will prevent confusion.

9. On lines 69-70, could you expand on how dog and human gut microbiomes are more similar? Do they have more taxa in common? I’m curious if this similarity is based on taxa or bacterial function.

10. The conjunction ‘but’ in the sentence on line 110-111 seems odd because it implies that just because chitosan is digestible by dogs its effects on the canine microbe should have been studied. I think this also undermines the great set-up you have for why you chose to study the association between the dog gut microbiome and chitosan/cricket meal. You might want to switch to another conjunction such as ‘and’.

Experimental design

11. The authors are inconsistent with how they filter their ASV table and if they use the rarefied table before an analysis. This seems odd to me. Could you clarify why you used different filtering thresholds across your analyses?

12. On line 134 you discuss how the dogs were housed. Can you clarify if these individual pens are all in the same facility? This may be relevant for future reproduction of your results given facility effects seen in other gut microbiome studies.

13. On line 146, can you include how long after the diet switch the initial fecal sample was taken? Including a lack of acclimation time as a caveat in your discussion was a great idea, but I also think that is why knowing the timeline of just how soon after the diet switch was made is important.

Validity of the findings

14. Consider using another description in the title other than “supports a healthy microbiome”. This language implies a cause and effect relationship that the study did not test. While there are ASVs increasing or decreasing that have been known to have health benefits in other studies, if these ASVs actually provided health benefits in this study’s dogs was not directly tested. Additionally, it may be best to avoid using the term ‘healthy microbiome’ as there is no consensus on what constitutes a healthy microbiome given that gut microbial composition among individuals that are considered healthy can vary greatly.

15. When discussing the lack of difference between the Bacteroidetes:Firmicutes (B:F) ratio among dogs fed different diets, consider modifying the sentence that states the lack means ‘cricket diets are likely not obesogenic relative to standard diets’. This seems strongly worded to me given that we are still learning more about the relationship between obesity and the gut microbiome and the B:F ratio is not the end-all be-all to tell you if a microbiome is obesogenic or not. In fact, more current evidence suggests that the B:F ratio and obesity connection does not hold up in robust meta-analyses. This could be one of the reasons the B:F ratio is not different if you suspect one of the diets plays a role in obesity. Here is a reference that discusses this: https://mbio.asm.org/content/7/4/e01018-16


16. On line 319, you write that the functional impact of ASVs with low abundance was “likely minimal”. Can you provide at least a reference to justify this statement? Especially because there are known examples of relatively low abundant bacteria that make important contributions to the function of the overall bacterial community.

17. On line 348, stating that a decrease in abundance of Bacterodetes “is likely” beneficial to health is strongly worded. Consider switching that language to “may be beneficial to health”.

18. Line 51 in the abstract states certain ASVs are “predicted to be beneficial to gut health”. Again, this seems strongly worded. You are not really ‘predicting’ that they are beneficial gut health, but you are mentioning it is a possibility given other literature. I would change this to “are associated with gut health benefits”.

19. On line 339 you write “the overall increase of Catenibacterium in dogs consuming cricket was health-promoting”. I would switch “was” to “may be” because you did not directly test if it was health promoting in this study.

Additional comments

I enjoyed reading this manuscript by Jarrett et al. that reports specific ASV abundance changes in the dog gut microbiome after consumption of cricket meal. Of particular interest are the dose-response changes they observed in some ASVs that showcase the link between cricket meal and abundance changes in certain ASVs. Overall, the manuscript is well written and thorough. However, I do have some recommendations that should improve clarity for the reader and ensure the authors do not overstate their work. Most notably, the authors should consider eliminating the current implications made by the wording of the title.

Reviewer 2 ·

Basic reporting

This manuscript is well written and concise. More importantly, the level of detail reported is sufficient to assess the strength of their conclusions and to repeat their work. Literature review was sufficient and the majority of relevant studies in the area were cited. It would be nice to have see the characteristics (age/weight) of individual animals as well as to have final weight, food intake, and evidence of any gastrointestinal issues (diarrhea, constipation, or gas) reported.

Experimental design

This study wanted to establish the potential for inclusion of cricket as a novel and sustainable source of protein and fiber in animal diets. Towards this end, they conducted a 4-arm study with varying amounts of cricket included in the diet and assessed changes in the microbiota. As mentioned in the previous section, a more detailed reporting of the animal characteristics would have been helpful to establishing this. However, the design and analysis approach of the microbiota is solid and provides useful information- particularly that including even large amounts of cricket is unlikely to be disruptive of the microbiota.

Validity of the findings

The authors do not try to oversell their results and were very transparent in their analysis approach and interpretation. Very well done!

---

## Round 0.2 · Minor Revisions

I'm satisfied that the revised manuscript has dealt with the comments raised by the two referees during peer review. I find the interpretation of the results is fair and balanced, including the caveats and directions for future research. I raise the following minor issues that I request some more details on:

- Please could you explain why the species of the cricket used in the study has changed since the first submission

- Related to the above, could you provide more detail in the manuscript on where the cricket meal and/or the prepared diets came from?

- You mention adjusting food volume to ensure steady mass, but only provide raw mass data and no statistics in the manuscript to show that mass remained constant, though I notice such statistics are in your ethical paperwork.

- Related to the above, you do not mention in the ethical paperwork that you were testing other health parameters such as RBC and haematocrit. Are these data being withheld for another publication? For example, if this paper were to be used to suggest that including cricket meal in dog food can support a healthy gut microbiome diversity comparable to 'regular' diets, I would expect any information relevant to any potential upper limit for the inclusion of cricket protein to be included here. Likewise, it would seem sensible to include such protocols in the methods of the study design.

- Strange font behaviour lines 228-233 of the reviewing pdf

- Same for line 258, font is in grey.

- Results reporting: please keep p values to 2-3 dp

---

## Round 0.3 · accepted · Accept

Many thanks for addressing the clarifications raised from the previous submission. I am now delighted to accept your manuscript.